# Biocide Tolerance and Impact of Sanitizer Concentrations on the Antibiotic Resistance of Bacteria Originating from Cheese

**DOI:** 10.3390/foods12213937

**Published:** 2023-10-27

**Authors:** Éva György, Károly Arnold Unguran, Éva Laslo

**Affiliations:** Department of Food Science, Faculty of Economics, Socio-Human Sciences and Engineering, Sapientia Hungarian University of Transylvania, 530104 Miercurea Ciuc, Romania; ungurankarolyarnold@uni.sapientia.ro (K.A.U.); lasloeva@uni.sapientia.ro (É.L.)

**Keywords:** cheese, bacterial diversity, antibiotic resistance, disinfectant, stress tolerance

## Abstract

In this study, we determined and identified the bacterial diversity of different types of artisanal and industrially produced cheese. The antibiotic (erythromycin, chloramphenicol, kanamycin, ampicillin, clindamycin, streptomycin, tetracycline, and gentamicin) and biocide (peracetic acid, sodium hypochlorite, and benzalkonium chloride) resistance of clinically relevant bacteria was determined as follows: *Staphylococcus aureus*, *Macrococcus caseolyticus*, *Bacillus* sp., *Kocuria varians*, *Escherichia coli*, *Enterococcus faecalis*, *Citrobacter freundii*, *Citrobacter pasteurii*, *Klebsiella oxytoca*, *Klebsiella michiganensis*, *Enterobacter* sp., *Enterobacter cloacae*, *Enterobacter sichuanensis*, *Raoultella ornithinolytica*, *Shigella flexneri*, and *Salmonella enterica.* Also, the effect of the sub-inhibitory concentration of three biocides on antibiotic resistance was determined. The microbiota of evaluated dairy products comprise diverse and heterogeneous groups of bacteria with respect to antibiotic and disinfectant tolerance. The results indicated that resistance was common in the case of ampicillin, chloramphenicol, erythromycin, and streptomycin. *Bacillus* sp. SCSSZT2/3, *Enterococcus faecalis* SRGT/1, *E. coli* SAT/1, *Raoultella ornithinolytica* MTT/5, and *S. aureus* SIJ/2 showed resistance to most antibiotics. The tested bacteria showed sensitivity to peracetic acid and a different level of tolerance to benzalkonium chloride and sodium hypochlorite. The inhibition zone diameter of antibiotics against *Enterococcus faecalis* SZT/2, *S. aureus* JS11, *E. coli* CSKO2, and *Kocuria varians* GRT/10 was affected only by the sub-inhibitory concentration of peracetic acid.

## 1. Introduction

Disinfection is used in the food industry to prevent microbial hazards. Biocides are utilized to control pathogens and spoilage bacteria, such as *Escherichia coli*, *Listeria monocytogenes*, *Staphylococcus aureus*, *Salmonella*, and *Pseudomonas aeruginosa* [1]. The effectiveness of biocides is determined by the type of microorganism. Biocides employed in the food industry contain active substances, such as chlorine, iodine, alcohols, quaternary ammonium compounds, hydrogen peroxide, silver, chlorhexidine, and triclosan. The biocide tolerance of foodborne pathogenic bacteria plays an important role in food safety, contributing to the survival of bacteria in adverse food environmental conditions. 

Foodborne pathogens undergo modifications in gene expression, thereby adapting to the changing environmental conditions and becoming tolerant to stress factors. This protective response increases the pathogenicity of bacterial strains and their viability during infections [2]. The alteration of membrane fluidity with modified lipid bilayer compositions, the inhibition of membrane enzymes, and enzymatic decomposition or DNA disruption contribute to their survival under food preservation and sanitation conditions [3,4]. Many pathogens and spoilage bacteria are able to adapt effectively to various stresses in their environment [5,6,7,8,9,10]. Biocides and other stress factors during food processing can affect the antibiotic resistance of various pathogenic bacteria (*Salmonella enteritidis*, *Escherichia coli* O157:H7, *Cronobacter sakazakii*, and *Listeria monocytogenes*) [11,12,13]. Biocide tolerance is linked to the emergence of antibiotic resistance. This is explained by the phenomenon of cross- and co-resistance in the cell based on different molecular mechanisms [4,14,15,16]. The presence of sub-lethal concentrations of a biocide results in stress responses in bacteria that produce changes in membrane permeability and membrane composition, thus affecting their efflux pumps. Via horizontal gene transfer, bacteria are also capable of developing new resistance and can survive technological processes [16]. There are several cases, mostly in biofilms, in which *Staphylococcus*, *Stenotrophomonas*, *Streptococcus*, *Pseudomonas*, *Acinetobacter*, and *Listeria* survive the disinfection process with increased pathogenicity [17]. Antibiotic-resistant bacteria have developed tolerance to biocides (benzalkonium chloride and chlorine-based), showing phenotypic and cell surface changes, as well as biofilm formation [10]. Peracetic acid has been shown to contribute to a higher reduction in pathogens on dairy-processing equipment surfaces and in biofilm tests [18]. Park et al. (2023) showed that at higher temperatures (30 °C), emetic *Bacillus cereus* strains became more virulent and tolerant to NaOCl and citric acid [19].

Cross-resistance to antimicrobials (chlorine and peracetic acid) was detected in *E. coli* and *Bacillus* sp. originating from food. Resistance to quaternary ammonium compounds was described in *Acinetobacter calcoaceticus*, *E. coli*, *Staphylococcus* sp., *Salmonella* sp., *Bacillus subtilis*, *Bacillus megaterium*, *Bacillus licheniformis*, and *Bacillus cereus*. It was observed that the latter bacteria were resistant to chlorine-releasing compounds [4]. Benzalkonium chloride was found to have an impact on antibiotic susceptibility in *E. coli* [20].

The identification and characterization of antibiotic-resistant microorganisms with biocide tolerance relative to cheese is essential in relation to food safety and technological perspectives. The utilization of raw milk in the production of cheese may contribute to microbiological hazards [21]. Inadequate technological parameters and fermentation conditions contribute to the growth and survival of these pathogen*s*, which cause the contamination of cheese [22]. Different types of raw milk cheese are a source of multidrug-resistant or methicillin-resistant and enterotoxin-producing *S. aureus* [23,24,25,26,27,28,29,30]. Raw and pasteurized milk cheese forms are linked to foodborne outbreaks, and the implicated bacteria are *L. monocytogenes*, *S. aureus*, *Salmonella* sp., Shiga-toxin-producing *E. coli*, *Campylobacter* sp., *Brucella* sp., *Shigella* sp., *Clostridium perfringens*, and *Bacillus cereus* [21,31,32,33,34,35].

Shiga-toxin-producing, enteropathogenic, and enterotoxigenic *E. coli* (ETEC O27:H20, EIEC O124:B17, EHEC O157, STEC, VTEC strains, O26, O103, O145, and O111) from different types of cheese was associated with severe symptoms and infections [21,34,36,37,38].

The aim of the present study is to evaluate the prevalence of antibiotic-resistant and biocide-tolerant bacterial strains from different artisanal and industrial cheese and to determine the impact of the most commonly used biocides on the antibiotic resistance of the identified bacteria.

## 2. Materials and Methods

### 2.1. Bacterial Isolates of Different Cheese Products

#### 2.1.1. Isolation of the Bacteria

The microbiological contamination of 21 different types and brands of cheese from the northeastern part of Transylvania (Romania) was tested for *Staphylococcus aureus*, *Escherichia coli*, *Salmonella* sp., and *Listeria monocytogenes* on selective media (microbiological criteria for cheese) using the direct plating method. In total, 25 g of each cheese sample was transferred into a 225 mL sterile physiological solution. A volume of 0.1 mL of each sample was spread on the selective agar medium. Incubation was performed at 37 °C for 48 h. The selective media used to isolate bacteria were as follows: Mannitol Salt Agar for *Staphylococcus aureus*; ChromoBio TBX media for *Escherichia coli* and coliforms; ChromoBio^®^ Salmonella Base and Brilliance ^TM^ Salmonella Agar Base for *Salmonella* sp.; and Listeria Identification Agar Base—Palcam and ChromoBio^®^ Listeria Plus for *Listeria monocytogenes* [39,40,41]. The cheese samples were local products obtained from stores and open-air public markets, including fresh sheep milk cheese, sheep whey cheese, fresh cow milk cheese, kashkaval cheese, feta-type cheese, and cow whey cheese. 

#### 2.1.2. Identification of Bacterial Isolates

The most representative bacteria strains developed on selective media with high cell count were isolated. Identification of the selected bacteria strains on the species level was obtained via 16S rDNA sequence analysis based on sequence similarity. Genomic DNA was isolated using Quick-DNA Fungal/Bacterial Miniprep Kit from Zymo Research. For the 16S ribosomal DNA amplification, the universal 27 (f 5′- AGAGTTTGATCMTGGCTCAG-3′) forward and 1492(5′TACGGYTACCTTGTTACGACTT-3′) reverse primers were used. Sequencing was carried out by Biomi KFT. (Gödöllő, Hungary). Sequences were handled using Chromas and MEGA software and compared with Basic Local Alignment Search Tool (BLAST) to known organisms within the National Center for Biotechnology Information (NCBI) nucleotide database for the isolates [42,43]. The criteria for species identification were 98.7% or higher similarity for the species level and 94.5% or higher similarity for the genus level [44].

### 2.2. Antimicrobial Susceptibility of Bacterial Strains

Antibiotic susceptibility testing of the identified bacteria was performed with the disk diffusion method. A total of eight different antibiotic disks containing the antibiotics erythromycin 10 μg (E), chloramphenicol 10 μg (C), kanamycin 30 μg (K), ampicillin 25 μg (AMP), clindamycin 10 μg (CD), streptomycin 25 μg (S), tetracycline 30 μg (TE), and gentamicin 50 μg (GEN) were used. The bacteria strains were classified as resistant, susceptible, or intermediate on the basis of inhibition zone diameter according to the guidelines of the European Committee on Antimicrobial Susceptibility Testing (EUCAST) [42,45]. Results are presented as means and standard deviations of three replicates. The multiple antibiotic resistance (MAR) index was determined as the ratio of the number of antibiotics to which the isolate exhibited resistance to the total number of antibiotics tested [46].

### 2.3. Antibacterial Effect of Biocides

The antibacterial effects of three commonly used biocides (peracetic acid, PAA; sodium hypochlorite, SHY; and pharmaceutically available benzalkonium-chloride-containing disinfectant, BZK) were evaluated with the agar diffusion method. The nutrient medium was inoculated on the surface with a 0.1 mL suspension of bacteria (10^8^ CFU/mL). In the center of all of the inoculated media, in the 6 mm diameter hole, 0.1 mL of the concentrated biocides (commercially available form PAA 15%, BZK 5 mg/mL, SHY 5%) was dropped. After incubation at 37 °C for 48 h, the results were read and expressed according to the size of the inhibition zone [47].

### 2.4. Bactericidal Concentration of Biocides in Nutrient Media 

To determine the bactericidal concentration of the biocides used for *Staphylococcus aureus* JS/11, *Escherichia coli* CSKO2, *Kocuria varians* GRT/10, and *Enterococcus faecalis* SZT/2, their growth was evaluated in the range of 50–400 parts per million (ppm) of PAA, 1–9 ppm BZK (1, 3, 4, 6, 8 ppm), and 1000–3750 ppm SHY (1000, 2000, 2500, 3000, 3500, and 3750). The growth inhibition was accepted when the optical density at 595 nm was less than 0.2 [48,49].

### 2.5. Effect of Sub-inhibitory Concentrations of Biocide on Antibiotic Resistance 

The effect of sub-inhibitory concentration of biocides on antibiotic resistance (adaptation of strains to sub-inhibitory concentration of biocides and in nutrient medium with biocide content) was realized according to Section 2.2. with modifications. The adaptation of the strains was evaluated by growth in the nutrient broth in the presence of 200 ppm (*Enterococcus faecalis* SZT/2, *Staphylococcus aureus* JS/11), 300 ppm (*Escherichia coli* CSKO2), and 50 ppm (*Korcuria varians* GRT/1) PAA for 24 h. The exposure to sub-inhibitory concentration was performed with 100 ppm PAA (*Enterococcus faecalis* SZT/2, *Staphylococcus aureus* JS/11), 150 ppm (*Escherichia coli* CSKO2) PAA, and 50 ppm (*Kocuria varians* GRT/10) PAA added to the media used for antimicrobial susceptibility testing [48,49].

### 2.6. Statistical Analysis 

Some results were analyzed using principal component analysis, and Spearman correlation was realized with the PAST software package. Spearman’s rank coefficients were used to determine the correlations between biocides and antimicrobial resistance; the significance level was considered as a *p*-value < 0.05.

## 3. Results and Discussion

### 3.1. Bacterial Isolates of Different Cheese Products

During this study, 21 different commercially available cheeses were evaluated microbiologically. Among the analyzed samples, there were 12 artisanal cheeses (eight made from sheep milk and four from cow milk), and 9 cheeses were produced on an industrial scale from cow milk. The most representative bacteria isolates developed on selective media with high colony count were isolated, and the pure cultures were examined. Identification of the selected 31 bacteria strains on the species/genus level was carried out via 16S rDNA sequence analysis (Table 1). The bacterial strains isolated from artisanal cheeses belong to different genera and are as follows*: Staphylococcus aureus*, *Macrococcus caseolyticus*, *Bacillus* sp., *Escherichia coli*, *Enterococcus faecalis*, *Citrobacter freundii*, *Klebsiella oxytoca*, *Klebsiella michiganensis*, *Enterobacter* sp., *Enterobacter sichuanensis*, *Shigella flexneri*, and *Salmonella enterica*. *Kocuria varians*, *Bacillus* sp., *Escherichia coli*, *Enterococcus faecalis*, *Citrobacter freundii*, *Klebsiella oxytoca*, *Enterobacter cloacae*, and *Raoultella ornithinolytica* were isolated from industrial cheeses. The most commonly encountered bacteria species was *Escherichia coli*, and the highest number of colonies was found in case of *Citrobacter freundii*. The selective medium supported the growth of other bacteria, not only the target bacteria.

Major pathogenic bacteria such as enterohemorrhagic *Escherichia coli* O157:H7, *Enterobacter* sp., *Shigella flexneri*, and *Salmonella enterica* were also found in artisanal cheeses.

The results suggest that different types of cheese purchased from different locations harbor the same bacterial species. Several studies report the occurrence of pathogenic antibiotic-resistant bacteria in different types of cheese worldwide. Other authors report that microbiological analysis shows similar results regarding the presence of different bacteria in cheeses. The majority of the analyzed types of cheese were contaminated with coliforms and *Staphylococcus* sp. as a result of poor food hygiene practices. *Staphylococcus aureus* and *M. caseolyticus* with pathogenicity islands containing genetic determinants of antibiotic-resistance (msr, SaRI*_msr_*, *mec*A, and *mec*D) are frequently detected in food samples [50,51]. Emerging multidrug-resistant pathogenic bacteria were detected in artisanal goat coalho cheeses as *Staphylococcus* sp., *Enterococcus* sp., *M. caseolyticus*, and *Enterobacter* sp. Genes responsible for antibiotic resistance *mecA* gene, *vanA*, and *vanB* were found in *Staphylococcus* sp., and *vanB* occurred in *Enterococcus* sp. [52]. Raw milk gouda was the source of *E. coli* O157:H7 that survived through the process line and ripening period, causing infections [53]. Fresh cheeses from Mexico were contaminated with *Salmonella*, *Listeria*, and *Escherichia coli O157:H7* [54].

Two analyzed cheeses were the source of spore-forming bacteria. The food poisoning emetic *Bacillus cereus* frequently contaminates different foods [55]. *Kocuria varians* is used as adjunct culture in the cheese-milk pre-ripening process, thereby contributing to the development of volatile compounds such as fatty acids, esters, and sulfur compounds, providing the flavor profile of Tetilla cheese [56]. The microbiological assay revealed the presence of *Citrobacter* sp. and *Klebsiella oxytoca* in different kashkaval-type cheeses, feta, and fresh cheese. *Enterobacteriaceae* contributes to the ripening of artisanal cheeses, despite the fact that it contains pathogenic members. It was found that during the initial ripening stages, *Citrobacter braakii*, *C. freundii*, *K. oxytoca*, and *Hafnia alvei* were present, and gas production was detected, causing early blowing in soft and semi-hard cheeses made from ewe milk [57]. 

*Enterobacter* sp. was detected in cow whey cheese, kashkaval cheese, and fresh cow cheese. The same bacteria from other sources showed resistance to amoxicillin clavunic acid, azomax, enoxacin, fosfomycin, fusidic acid, gentamycin, moxifloxacin, piperacillin, tazobactam, sulbactam, and sparfloxacin [58,59]. *Citrobacter pasteurii* and *Citrobacter braakii* are able to produce cicrocins with an antibacterial effect against closely related bacteria [60].

Poor sanitation conditions are a leading risk factor for gastroenteritis-causing *Shigella*. In fresh cow cheese, S. *flexneri* was detected. This bacterium causes endemic shigellosis and is transmitted via the fecal-oral route. Raw milk cheese from Peru containing *Shigella* caused gastroenteritis in immunosuppressed patients [61].

One of the kashkaval-type cheeses contained *Raoultella ornithinolytica*, which was identified in traditionally produced Turkish white cheese [62]. The number of infections associated with *R. ornithinolytica* has increased, but this bacterium is hard to differentiate from K*. oxytoca*, *Enterobacter aerogenes*, and *Raoultella* sp. [63]. 

### 3.2. Antimicrobial Susceptibility of Bacterial Strains

Resistance to antibiotics represents a threat worldwide. The 31 bacterial isolates originating from different types of cheese were tested for susceptibility to eight of the most frequently used antibiotics with the agar diffusion method. Seven bacterial strains showed resistance to at least four antibiotics (Table 2).

*Enterococcus faecalis* SRGT/1 was resistant to kanamycin, gentamicin, streptomycin, and clindamycin. *Raoultella ornithinolytica* MTT/5 showed resistance to erythromycin, tetracycline, streptomycin, and clindamycin. *Staphylococcus aureus* SIJ/2 exhibited resistance to ampicillin, tetracycline, erythromycin, and streptomycin. *Citrobacter freundii* SCSSZT2/1 was resistant to five antimicrobials. These bacteria strains showed no resistance to chloramphenicol, tetracycline, or gentamicin.

The broad range of resistance was detected in case of *Bacillus* sp. SCSSZT2/3, which was only susceptible to erythromycin and tetracycline. Zhai et al., 2023 [64], reported that *Bacillus cereus* strains isolated from milk harbor acquired resistance genes. The analyzed *Bacillus* sp. strains showed acquired resistance to clindamycin, as detected in our case. There are bacteria strains causing spoilage that harbor antibiotic resistance genes. *Escherichia coli* SAT/1 was resistant to ampicillin, clindamycin, kanamycin, and streptomycin. *Klebsiella michiganensis* SVJ/3 was resistant to gentamicin, streptomycin, ampicillin, and clindamycin.

Multidrug resistance in foodborne pathogens and spoilage bacteria has proved an increasing threat globally through the food chain. The multiple antibiotic resistance (MAR) index indicates that the bacteria originate from sources where antibiotics are used in large amounts [65]. The MAR index of the 31 tested bacterial strains ranged from 0.125 to 0.75, with maximum number of one bacterial strain resistant to six antimicrobials. A MAR index value higher than 0.2 indicates that the bacterial strains come from sources where antibiotics are frequently used [65]. The bacteria in our investigation are as follows: *Staphylococcus aureus* SIJ/2 (MAR = 0.5), *Citrobacter* sp. SCSJ/1 (MAR = 0.25), *Citrobacter freundii* SCSSZT2/1 (MAR = 0.625), *Citrobacter freundii* KSZT5 (MAR = 0.25), *Citrobacter pasteurii* SSZMH/1 (MAR = 0.25), *Bacillus cereus* SCSSZT2/3 (MAR = 0.625), *Escherichia coli* SAT/1 (MAR = 0.5), *Escherichia coli* TKO/1 (MAR = 0.375), *Enterobacter sichuanensis* SJT/8 (MAR = 0.25), *Enterobacter* sp. OAT/1 (MAR = 0.25), *Klebsiella michiganensis* SVJ/3 (MAR = 0.5), *Shigella flexneri* SVT/1 (MAR = 0.25), *Raoultella ornithinolytica* MTT/5 (MAR = 0.5), and *Enterococcus faecalis* SRGT/1 (MAR = 0.5). The current research shows the occurrence of various bacterial strains in different types of cheese. It was found that a high number of bacterial strains showed susceptibility to the major tested antibiotics, but it was also found that antibiotic-resistant strains show tolerance to the major biocides. Bacterial susceptibility to the tested antibiotics is not related to the type of bacteria source.

Several foodborne pathogens and spoilage bacteria have become resistant to antibiotics after different infection treatments. Akter et al., 2023 [66], highlighted that the *Enterococcus faecalis* genome sequence contained several antibiotic-resistant genes, including S12p^p^, gidBp^p^ (streptomycin), amp(S)a (beta-lactamases). The multidrug-resistant *R. ornithinolytica* is a growing threat, causing human infections. It was summarized that *R. ornithinolytica* isolates showed resistance to amoxicillin, to quinolones, and to aminoglycosides [63,67]. Ampicillin, aminoglycosides, and sulfonamides are commonly used to treat bovine mastitis. Clindamycin, tetracycline, and erythromycin are regularly detected in wastewater and cattle manure [64,68].

Principal component analysis (PCA) was used to assess the association between antibiotic-resistant and sensitive bacterial strains (Figure 1). The results of the multivariate data analysis provide an indication of the arrangement of strains according to their antibiotic sensitivity or resistance pattern. Among the principal components, F1 accounts for 31.673% of the cumulative variability, F2 for 24.09%, and F3 for 16.43%, explaining 72.19% of the total variability.

### 3.3. Antibacterial Effect of Biocides

Biocides are chemical formulations containing at least one active substance. These chemicals with an antimicrobial effect are also called disinfectants or sanitizers. The antimicrobial potential of PAA, BZK, and SHY was determined with the agar diffusion method (measuring the inhibition zone diameter). The concentrated PAA completely inhibited all bacterial strains. The antimicrobial activity of the biocides showed differences among the bacterial strains (Figure 2 and Figure 3). The highest efficiency (inhibition zone diameter > 20 mm) was observed for BZK against the strains *Staphylococcus aureus* JS11 and SIJ/2, *Macrococcus caseolyticus* OIJ/2 and *Macrococcus* sp. SAT/4, *Bacillus* sp*. SCSSZT2/3*, *Escherichia coli* JS10, *Enterococcus faecalis* SZT/2, *Klebsiella oxytoca* SSZMJ3/4 and OIJ/6, *Klebsiella michiganensis* SVJ/3, *Enterobacter* sp. OAT/1, and *Salmonella enterica* SSZMJ2/4. In contrast, the SHY was more efficient against the tested bacterial strains, and *Macrococcus caseolyticus* OIJ/2 and *Salmonella enterica* SSZMJ2/4 showed tolerance to SHY.

The strength of the relationship between the inhibition zone size of antibiotics and biocides or two antibiotics was also determined (Figure 4). Spearman’s rank correlation coefficient (r_s_) was used to determine the correlations between antibiotics and biocides based on inhibition zone sizes (mm) (Figure 4). 

The Spearman correlation showed statistically significant negative correlations (at a significance level of *p* < 0.05) between SHY and AMP (r_s_ = −0.48), C (r_s_ = −0.41), E (r_s_ = −0.36), S (r_s_ = −0.35), and CD (r_s_ = −0.37), whereas a positive correlation was found between BZK (r_s_ = −0.36) and S, as well as between BZK and the antibiotics E (r_s_ = 0.22), AMP (r_s_ = 0.23), and CD (r_s_ = 0.11). 

Based on the r_s_ values, the correlation was moderate (0.4 ≤ rs < 0.7) or weak (0.1 ≤ rs < 0.4) [69,70]. A significant positive moderate correlation (*p* < 0.05) was found between the antibiotics CD and K (r_s_ = 0.43), E (r_s_ = 0.56), and G (r_s_ = 0.52), between S and AMP (r_s_ = 0.4), K (r_s_ = 0.67), TE (r_s_ = 0.61), and between C and AMP (r_s_ = 0.6), while a weak correlation was found between G and AMP (r_s_ = 0.38).

This analysis reveals that the antimicrobial effects of some antibiotics are similar.

The outcomes of this research show that the sensitivity of bacteria strains to biocides from different cheese production environments was distinct. The obtained results could be explained by the fact that the used biocides exert selective pressure on the bacterial strains, favoring the survival of resistant strains that become tolerant to the used chemicals. The safety and quality of disinfection is related to the type of applied chemicals. Sodium hypochlorite is widely used as a strong oxidizing sanitizer in the food industry. The bactericidal mechanism of this disinfectant is based on the destruction of membranes, enzymes, and DNA. There are several factors that determine the efficacy of disinfectants, such as pH, contact time, temperature, concentration, the microbe and its growth conditions, and interfering substances. The results show that *Raoultella ornithinolytica* MTT/5 was highly sensitive to SHY. Tantasuttikul and Mahakarnchanakul, 2019 [69], reported that *Raoultella* sp. was completely killed at a concentration of 0.5 mg/L SHY for 5 min, but lower concentrations required longer contact time. The use of 0.02% NaClO (20 mg/L) for 15 min contributed to ≥5 log10 reductions in *Acinetobacter baumannii*, *Acinetobacter pittii*, *P. aeruginosa*, *Klebsiella pneumoniae*, and *Klebsiella oxytoca* [70]. In contrast to the present results, the *Klebsiella* strains showed large inhibition zones in the presence of 5% SHY. Bacterial species from sheep’s milk cheeses, in contrast to those from cow’s milk cheeses, showed larger inhibition zone diameters against BZK. 

It was observed that *E. coli* developed tolerance to various disinfectants as a result of acid resistance systems. This anti-acid stress system in *E. coli* is supported by the presence of different amino acids that consume intracellular protons through decarboxylation [49].

Briseño-Marroquín et al., (2022) [71], observed that PAA (0.5–2.0%) exerted large inhibition zones against *E. faecalis* and relatively small zones against *Parvimonas micra.* With decreasing concentration of 5.0, 3.0, and 1.0% of NaOCl, the diameters of the inhibition zones against *E. faecalis* also increased. 

PPA is a highly oxidizing biocide that exerts a bactericidal effect on bacterial cells. The commercially available concentrated PAA resulted in total inhibition of the bacteria tested. Cell wall permeability is affected by disruption of sulfhydryl and sulfur bonds [72]. It was observed that PAA treatment of various bacteria, especially in *Staphylococcus aureus SA1*, caused damage to cell membrane. Membrane integrity was affected as a result of enzyme inactivation and changes in membrane potential [73].

BZK belongs to the lytic cationic biocides that interact with the negatively charged phospholipid bilayer of bacterial cell membranes via the positively charged portion. This leads to membrane destabilization, causing cell lysis, leaking cell components, and contributing to cell death [74]. Some of the tested antibiotic-resistant bacteria from cheeses from the northeastern part of Transylvania show tolerance to BZK. 

### 3.4. Bactericidal Concentration of Biocides in Nutrient Media 

The bactericidal concentrations of the three biocides used for *Staphylococcus aureus* JS11, *Escherichia coli* CSKO2, *Kocuria varians* GRT/10, and *Enterococcus faecalis* SZT/2, are shown in Table 3.

The bactericidal concentration that prevented the survival of the four bacteria tested varied in the range of 50–400 ppm PAA. These levels correspond to those used in the food industry (50–2000 ppm) [75]. The MIC concentration was equal to or less than 300 ppm. Growth inhibition was accepted at an OD at 595 nm lower than 0.2 in all cases. The maximum concentration of PPA that allowed the growth of *Staphylococcus aureus* JS11 was 200 ppm, and the bactericidal concentration was 210 ppm. The growth of *Escherichia coli* CSKO2 was restricted in the presence of 400 ppm PAA. The concentration of PAA at which growth of this bacterium was observed was 400 ppm, in the test tube assay. The minimum inhibitory concentration was 340 ppm. The maximum concentration of PAA that allowed the growth of *Kocuria varians* GRT/10 was 50 ppm. The bactericidal concentration was 90 ppm. In the case of *Enterococcus faecalis* SZT/2, the maximum concentration of PPA that allowed the growth of *Staphylococcus aureus* JS11 was 200 ppm. The MIC concentration in the growth assay was less than 300 ppm. The bactericidal concentration was 290 ppm.

The growth of *Escherichia coli* CSKO2 was restricted in the presence of 6 ppm BZK, whereas growth was supported in the presence of the 4 ppm concentration. *Kocuria varians* GRT/10 was able to grow only in the presence of 1 ppm BZK, whereas the 3 ppm concentration inhibited the growth of this bacterium. It should be noted that BZK was used as a pharmaceutical formulation containing chlorhexidine digluconate in addition to BZK. The results of MICs are above the limited concentrations in food processing plants of 200–400 ppm but are similar to the results of other studies [76,77]. For the other two bacterial strains, the used concentrations of BZK limited the growth.

The growth of *Escherichia coli* CSKO2 was restricted in the presence of 3000 ppm SHY. The concentration that allowed bacterial growth was 2500 ppm. *Staphylococcus aureus* JS11 was inhibited at 2000 ppm SHY, and growth was supported at 1000 ppm concentration. 

### 3.5. Effect of Sub-inhibitory Concentrations of Biocide on Antibiotic Resistance 

After exposure (*Staphylococcus aureus* JS11 100) and adaptation (*Staphylococcus aureus* JS11 200) to sub-inhibitory concentrations of PAA, differences in inhibition zone values were detected for E, TE, C, GEN, and AMP (Figure 5). After exposure (*Escherichia coli* CSKO2 150) and adaptation (*Escherichia coli* CSKO2 300) to sub-inhibitory concentrations of PAA, differences in inhibition zone values were detected in the case of C10. After exposure (*Kocuria varians* GRT/10 50) and adaptation (*Kocuria varians* GRT/10 100) to sub-inhibitory concentrations of PAA, differences in the inhibition zone values were detected in the case of AMP, E, and CD (Figure 5). Exposure or adaptation to the other two biocides had little or no effect on inhibition zone diameters. After exposure (*Enterococcus faecalis* SZT/2 100) and adaptation (*Enterococcus faecalis* SZT/2 200) to sub-inhibitory concentrations of PAA, differences in the inhibition zone values (Figure 6) were observed for AMP, C, and S.

It is considered that the spread of antibiotic-resistant bacteria is a consequence of the overuse of disinfectants through some complex regulatory pathways and genetic modifications [78]. In the food processing industry, reducing the amount of disinfectant that contributes to the spread of resistant bacterial strains is not allowed. The use of biocides at sub-lethal concentrations can increase bacterial resistance to antibiotics and disinfectants. The decrease in susceptibility of the most important antibiotics for humans such as CIP, CTX, FOX, and STR is the result of the efflux pump [48].

The antibiotic susceptibility of bacterial strains to different classes of antimicrobials was altered after exposure to biocides. Bacterial adaptation to sanitizers is related to a broad spectrum of mechanisms such as increased expression of nonspecific-efflux pumps and alteration of cell membrane permeability. Exposure of *Salmonella enterica* to SHZ, BZK, PAA, and trisodium phosphate altered the antibiotic resistance patterns of the bacterium. *Cronobacter sakazakii* ATCC 29544 and *Yersinia enterocolitica* ATCC 9610 showed an acquired tolerance to SHZ, BZK, and PAA and an increase in antibiotic resistance after exposure to sub-inhibitory concentrations of biocides. The growth parameters of the studied strains were not affected by the presence of biocides at sub-inhibitory concentrations [48]. Kumar et al., 2023 [49], reported cross-tolerance to PAA in *E. coli* O157 and non-O157 strains. In *Pseudomonas* sp. strains, the sub-inhibitory concentrations of NaClO contributed to the increase in the minimum inhibitory concentration (MIC) of antibiotics such as piperacillin–tazobactam, ciprofloxacin, gentamicin, meropenem, amikacin, ceftazidime, and colistin [70].

## 4. Conclusions

Our study contributes to the microbiological analysis of commercially available cheeses from the northeastern part of Transylvania, which harbor coliforms and staphylococci, besides other bacteria. The sampled cheeses were the source of antibiotic-resistant bacteria with biocide tolerance, such as *Bacillus* sp. SCSSZT2/3, *E. coli* SAT/1, *Raoultella ornithinolytica* MTT/5 and *Enterococcus faecalis* SZT/2. The microbiological results reveal poor sanitation and hygienic conditions, which should be taken into account when biocides are applied. The analyzed cheeses were contaminated with pathogenic bacteria such as *Salmonella enterica* SSZMJ2/4 and *Shigella flexneri* SVT/1 with tolerance to BZK and SHY and sensitivity to concentrated PAA. The research results highlight that the presence of sub inhibitory concentrations of commonly used biocides can have an impact on the antibiotic resistance of certain bacteria. The susceptibility of *Staphylococcus aureus* JS11, *Kocuria varians* GRT/10, and *Enterococcus faecalis* SZT/2 to some antibiotics was slightly affected. These findings are of great concern for public health safety.

## Figures and Tables

**Figure 1 foods-12-03937-f001:**
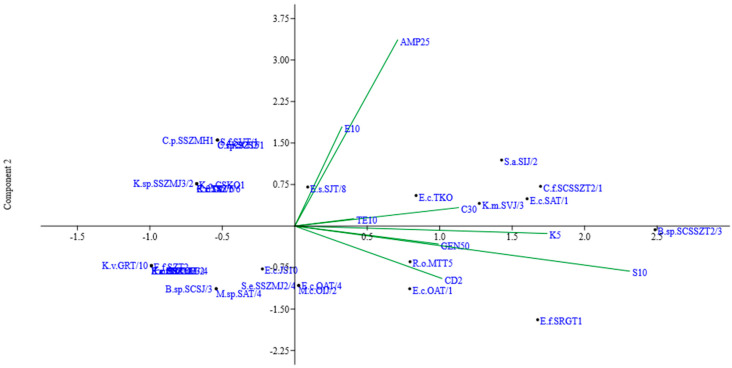
PCA of antibiotic-resistant and sensitive bacterial strains isolated from different cheeses.

**Figure 2 foods-12-03937-f002:**
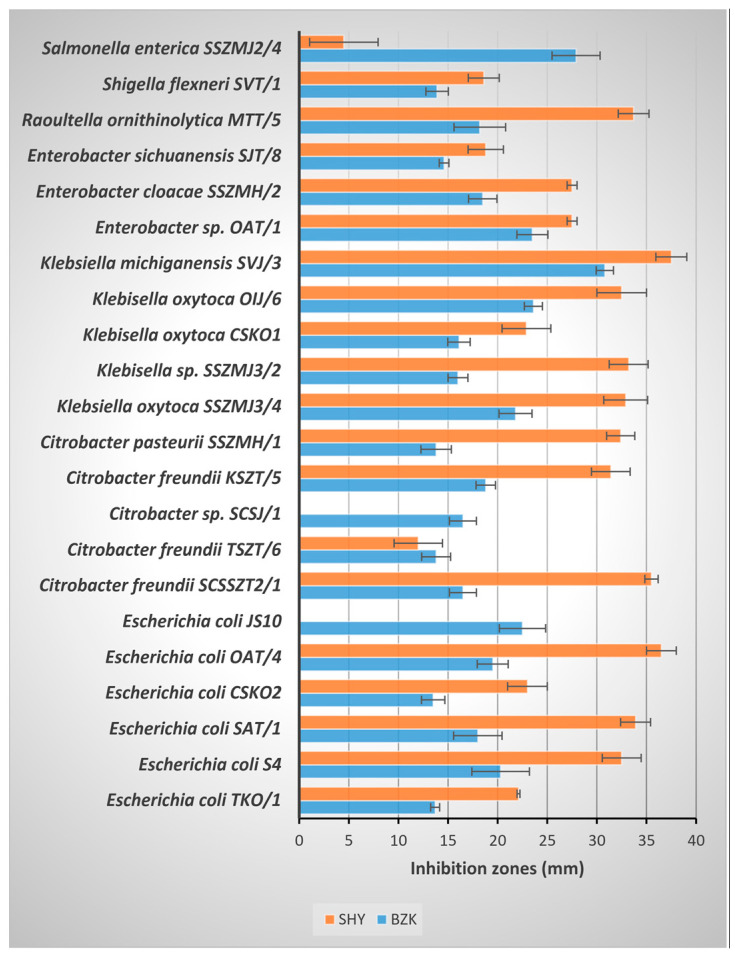
The antibacterial effect of commercially available BZK and SHY (inhibition zones in mm), on Gram-negative bacteria.

**Figure 3 foods-12-03937-f003:**
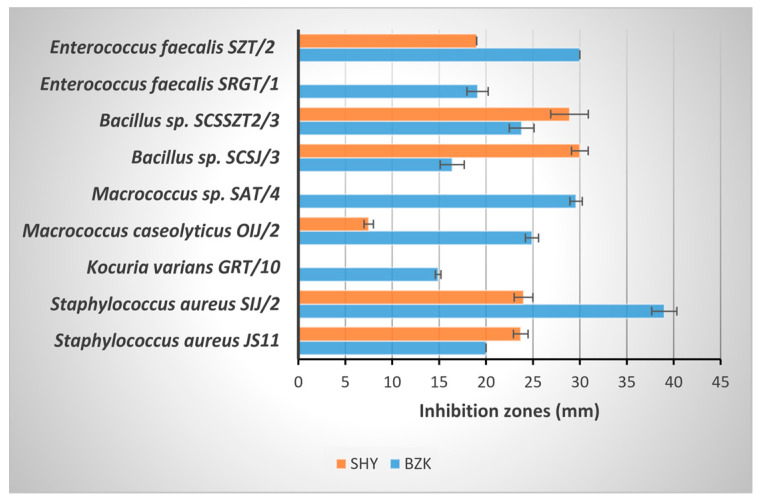
The antibacterial effect of commercially available BZK and SHY (inhibition zones in mm), on Gram-positive bacteria.

**Figure 4 foods-12-03937-f004:**
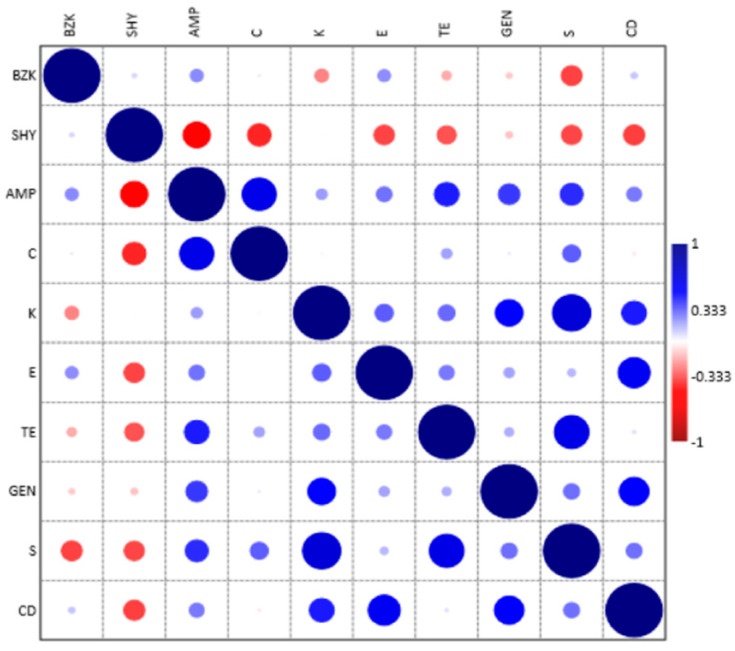
Spearman correlation coefficient graphic between inhibition zone size (mm) of antibiotics and antibacterial activity of biocides. Color-coded values range from −1 = negative correlation (red) to 1 = positive correlation (blue); color intensity and bubble size co-vary with the size of the data points.

**Figure 5 foods-12-03937-f005:**
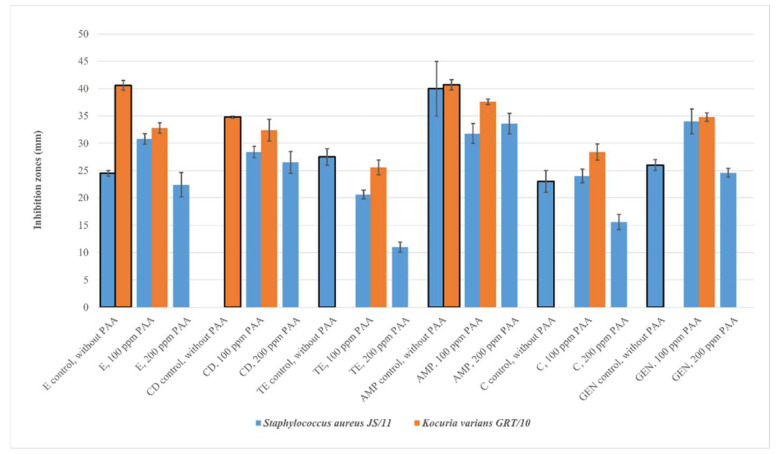
Inhibition zone diameter (mm) of antibiotics against *Staphylococcus aureus* JS11 and *Kocuria varians* GRT/10 adapted or exposed to sub-inhibitory concentrations of biocides and the inhibition zones without PAA pretreatment.

**Figure 6 foods-12-03937-f006:**
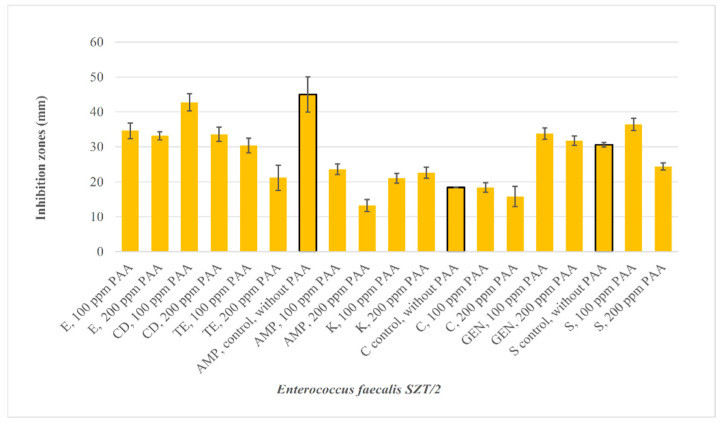
Inhibition zone diameter (mm) of antibiotics against *Enterococcus faecalis* SZT/2 adapted or exposed to sub-inhibitory concentrations of biocides and the inhibition zones without PAA pretreatment.

**Table 1 foods-12-03937-t001:** The source and identification of the bacterial strains isolated from the different cheeses.

Source of Isolation	Isolation Medium	Number of Colonies CFU/g	Identified Closely Related Species Based on 16S rDNA	Sequence Similarity %
Cow whey cheese	Brilliance ^TM^ Salmonella Agar Base	5.2 × 10^3^	*Escherichia coli*	99.05
Cow whey cheese	ChromoBio TBX	5 × 10^3^	*Enterobacter* sp.	98.54
Feta-type cheese	ChromoBio TBX	4.8 × 10^2^	*Escherichia coli*	99.8
Feta-type cheese	Listeria Identification Agar Base—Palcam	1.5 × 10^2^	*Citrobacter freundii*	99.52
Fresh cow cheese	Listeria Identification Agar Base—Palcam	1.4 × 10^2^	*Enterobacter sichuanensis*	99
Fresh cow cheese	ChromoBio TBX	2.3 × 10^2^	*Shigella flexneri*	99
Fresh cow milk cheese	ChromoBio^®^ Listeria Plus	1.6 × 10^2^	*Macrococcus* sp.	96.53
Fresh cow milk cheese	ChromoBio TBX	9.2 × 10^2^	*Escherichia coli*	99.33
Fresh cow milk cheese	Mannitol Salt Agar	1.6 × 10^2^	*Enterococcus faecalis*	99.42
Fresh sheep milk cheese	Mannitol Salt Agar	2.5 × 10^3^	*Staphylococcus aureus*	99.71
Fresh sheep milk cheese	Mannitol Salt Agar	5 × 10^2^	*Staphylococcus aureus*	100
Fresh sheep milk cheese	Brilliance ^TM^ Salmonella Agar Base	4.5 × 10^2^	*Bacillus* sp.	96.15
Fresh sheep milk cheese	ChromoBio TBX	1.7 × 10^2^	*Escherichia coli O157:H7*	100
Fresh sheep milk cheese	ChromoBio TBX	1.59 × 10^3^	*Escherichia coli*	99.71
Fresh sheep milk cheese	ChromoBio TBX	1.5 × 10^4^	*Citrobacter* sp.	98.57
Fresh sheep milk cheese	ChromoBio^®^ Salmonella Base	5.6 × 10^2^	*Klebsiella oxytoca*	99.81
Fresh sheep milk cheese	ChromoBio TBX	9 × 10	*Klebsiella* sp.	92.53
Fresh sheep milk cheese	ChromoBio^®^ Listeria Plus	2.4 × 10^2^	*Klebsiella michiganensis*	99
Fresh sheep milk cheese	ChromoBio^®^ Listeria Plus	2.1 × 10^2^	*Salmonella enterica*	99.04
Kashkaval cheese	Listeria Identification Agar Base—Palcam	3 × 10	*Kocuria varians*	99.04
Kashkaval cheese	Brilliance ^TM^ Salmonella Agar Base	2.1 × 10^4^	*Bacillus* sp.	97.64
Kashkaval cheese	ChromoBio TBX	5.8 × 10^2^	*Escherichia coli*	99.09
Kashkaval cheese	Listeria Identification Agar Base—Palcam	2.5 × 10^2^	*Enterococcus faecalis*	99.81
Kashkaval cheese	ChromoBio TBX	5 × 10^4^	*Citrobacter freundii*	99.51
Kashkaval cheese	Brilliance ^TM^ Salmonella Agar Base	2.8 × 10^2^	*Citrobacter freundii*	98.89
Kashkaval cheese	ChromoBio TBX	1.47 × 10^3^	*Citrobacter pasteurii*	99.34
Kashkaval cheese	ChromoBio TBX	5.8 × 10^2^	*Klebsiella oxytoca*	99.31
Kashkaval cheese	ChromoBio TBX	1.11 × 10^3^	*Enterobacter cloacae*	99.71
Kashkaval cheese	Brilliance ^TM^ Salmonella Agar Base	4 × 10	*Raoultella ornithinolytica*	99.17
Sheep whey cheese	Mannitol Salt Agar	3 × 10^2^	*Macrococcus caseolyticus*	99
Sheep whey cheese	ChromoBio^®^ Listeria Plus	1.24 × 10^3^	*Klebsiella oxytoca*	100

**Table 2 foods-12-03937-t002:** Antibiotic resistance pattern of bacterial strains.

Bacteria Strain	Antibiotic Resistance	Bacteria Strain	Antibiotic Resistance
*Staphylococcus aureus* SIJ/2	AMP, E, TE, S	*Citrobacter* sp. SCSJ/1	AMP, E
*Macrococcus caseolyticus* OIJ/2	S	*Citrobacter freundii* KSZT/5	AMP, E
*Macrococcus* sp. SAT/4	CD	*Citrobacter pasteurii* SSZMH/1	AMP, E
*Bacillus* sp. SCSJ/3	CD	*Klebsiella* sp. SSZMJ3/2	AMP
*Bacillus* sp. SCSSZT2/3	AMP, C, K, GEN, S, CD	*Klebsiella oxytoca* CSKO1	AMP
*Escherichia coli O157:H7* S4	AMP	*Klebsiella oxytoca* OIJ/6	AMP
*Escherichia coli* TKO/1	AMP, C, S	*Klebsiella michiganensis* SVJ/3	AMP, C, GEN, S
*Escherichia coli* SAT/1	AMP, C, K, S	*Enterobacter* sp. OAT/1	K, S
*Escherichia coli* OAT/4	S	*Enterobacter sichuanensis* SJT/8	AMP, K
*Escherichia coli* JS10	K	*Raoultella ornithinolytica* MTT/5	E, TE, S, CD
*Enterococcus faecalis* SRGT/1	K, GEN, S, CD	*Shigella flexneri* SVT/1	AMP, E
*Citrobacter freundii* SCSSZT2/1	AMP, K, E, S, CD	*Salmonella enterica* SSZMJ2/4	S
*Citrobacter freundii* TSZT/6	AMP		

**Table 3 foods-12-03937-t003:** Lowest biocide concentrations permitting bacterial growth.

Microorganism	Biocide	Lowest Concentration Permitting Bacterial Growth ppm
*Staphylococcus aureus* JS11	PAA	200
	SHY	1000
*Escherichia coli* CSKO2	PAA	300
	SHY	2500
	BZK	4
*Kocuria varians* GRT/10	PAA	50
	SHY	-
	BZK	1
*Enterococcus faecalis* SZT/2	PAA	200
	SHY	-

## Data Availability

The article contains all the relevant data. The original contributions presented in the study are included in the article; further inquiries can be directed to the corresponding author.

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
