# Peer review of "Biocide Tolerance and Impact of Sanitizer Concentrations on the Antibiotic Resistance of Bacteria Originating from Cheese"

_foods, 2023, doi:10.3390/foods12213937_

Round 1

Reviewer 1 Report

In this study, the authors isolated several bacterial species from various types of cheese and assessed their susceptibility to multiple antibiotics and common biocides. They also investigated whether sub-lethal concentrations of biocides could influence bacterial antibiotic susceptibility. While the research addresses the crucial issue of antibiotic resistance in foodborne bacteria, there are several aspects of the methods and results that could benefit from clarification and improvement to enhance readability and reliability:

Methods:

1. It would be helpful to specify the source of the cheese used in this study and discuss the extent to which the findings can be generalized to other cheese sources or types.

2. Information regarding the number of colonies isolated and sequenced from each selective plate should be provided to give a better understanding of the study's scope.

3. Clarify the unit of "ppm" and provide a context for how the concentrations of biocides used in this work compare to standard sterilization protocols in food factories.

4. Describe the methodology used to evaluate the abundance of bacteria in cheese, particularly concerning the statement in Line 173 about Citrobacter freundii being the most abundant bacterium. Supporting data or methods should be included.

Results:

1. Address and explain the presence of unexpected species on selective agar shown in Table 1, such as Enterococcus on Mannitol Salt Agar, Salmonella on Listeria-selective agar, and Bacillus on Salmonella-selective agar.

2. Section 3.2 references antibiotic resistance results, but no data is presented in the manuscript.

3. In Figure 2, provide information about the concentrations of SHY and BZK used. And how does the inhibition zone diameter correlate with the MIC of the biocides?

4. In Figure 4, which illustrates the inhibition zone against bacteria pre-treated with PAA, the figure should include a control without PAA pre-treatment. This would help indicate whether PAA has an actual effect on antibiotic susceptibility.

Proofreading is needed for this manuscript to fix the grammar issues. The authors may consider separating some of the long sentences to multiple short sentences to improve the readability. Some examples:

Line 32, typo “control".

Lines 63-66, this sentence is hard to understand.

Lines 156-158, delete this paragraph.

Line 282, “this bacteria growth” to “the growth of this bacterium”

Reviewer 2 Report

 Biocide tolerance and impact of sanitizer concentration on antibiotic resistance of bacteria originated from artisanal and industrial cheese is observational study finding antibiotic resistant and biocide tolerant microbes in commercial cheeses, with evidence that biocide exposure may increase antibiotic resistance in some bacteria. The small-scale study points to potential food safety risks. The concept of the study is good and overall quality of the paper but have some major issues which need to be addressed before accepting the paper.

·       In line 93 staph aureus is repetitive

·       In methods, it need to be clarify the sample was directly plated on selective media or it was cultured before and then for further identification selective media was used.

·       Why only selective media was used to isolate specific bacteria like E. coli and Staphylococcus. Broader microbiological testing may reveal other relevant microbes.

·       The antibiotic susceptibility testing was checked against 8 antibiotics on which bases these antibiotics were selected.

·       Line 126 lack evidence.

·       Line 154 re write it, currently it is unclear.

·       Line 168-173, here authors reported different bacterial isolates but in method it was declared, looking for three

·       The method and results need to be align.

·       Table 2, text is overlapping.

·       Fig 4, I will suggest to put bacterial isolates on x axis for better visualization

·       The study was limited to analysing cheeses available commercially in the authors' location. The results may not be generalizable to cheeses from other regions/countries.

Reviewer 3 Report

The manuscript presents an interesting study on the resistance of foodborne bacteria to biocides and antibiotics. It needs, however, considerable improvement. Several important methodological issues are present, and they affect the validity of some of the conclusions of the study. I have serious qualms about the validity of correlating inhibition diameters in the kind of test the authors used (agar diffusion). Measurement units are missing in tables and figures. The number of replicates is not clearly indicated. English is poor, with several unintelligible sentences, and part of the text is not visible, because a table was superimposed on it. Most of the manuscript consists of long, tedious descriptions of what is presented in the tables and figures. One of the figures is too large. Parts of the text of MDPI’s template have been left in the manuscript and should be deleted. For the sake of conciseness and readability, the manuscript would benefit from presenting one single Results and Discussion instead of a separate section for each of these. A thorough revision of the text by a native English speaker is mandatory – the text is rife with grammar faults. Names of bacterial genera and species are often not the valid ones. The authors need to consult the List of Prokaryotic Names with Standing in Nomenclature to verify whether they correct names are being used, and to correct those that are no longer valid. Detailed comments are given below. Please bear in mind that English must be revised throughout the manuscript and not only the examples provided.

Title: “… originated from artisanal and industrial cheese”; when reading this title, the author is guided to expect a comparison between bacteria from artisanal and those from industrial cheese. That is not, however, the direction towards which the authors steer their manuscript. Therefore, either the title should read only “…from cheese”, or a comparative analysis of the results of resistance in bacteria form artisanal and those from industrial cheese should be added.

Abstract: Should be rewritten to accent the most important results of the study – and to improve English.

Line 17 – 18: “The evaluated dairy products are comprised of 17 diverse and heterogeneous groups of bacteria with antibiotic and disinfectant tolerance”. Please rephrase for clarity and grammatical correctness. Do you mean “the microbiota of the evaluated dairy products”?

Line 22 – Please correct “para acetic” to “peracetic”.

Lines 25 – 26: Please rephrase for clarity and grammatical correctness. In its present form, it is not possible to understand what the authors mean with this sentence.

Lines 33 – 35: This sentence also needs to be rewritten in a clearer language. Please make a better link between biodiversity and the possibility of occurrence of resistant bacteria, keeping the sentence succinct.

Lines 35 – 38: “Biocides containing active substances are chlorine…”. Please rewrite to make the sentence grammatically correct and clear to the readers. Do you mean to say that “Biocides employed in the food industry contain active substances such as chlorine…”? If yes, please correct accordingly.

Line 41 – “pathogeny” – the correct term in this context is “pathogenicity” (or “virulence”). Please correct accordingly.

Line 44 – “survival of food preservation and sanitation conditions” – do you mean “survival to food preservation and sanitation conditions”? Please correct accordingly.

Line 45 – “prevalent emergence”? This expression does not make grammatical sense. Please correct.

Line 50 – replace “co-and” with “co- and”.

Line 51 – a full stop is missing after [4,14–16].

Line 52 – “produces” instead of “produce”

Line 53 “–“ instead of "-“

Line 55 – Revise punctuation; it is not correctly used.

Line 61 – B. cereus – it is the first time you mention this species; please give its name in full.

Line 64 – A. calcoaceticus – it is the first time you mention this species; please give its name in full.

Lines 63 – 67: This paragraph starts with “Cross-resistance was detected in food originated E. coli,…”. However, the text is only about resistance of several bacteria to one type of biocide – it does not demonstrate the link between resistance to one type of compound and that to other types of compounds. Please give examples of cross-resistance in this paragraph.

Lines 75 – 83: The text in these paragraphs needs to be revised for conciseness, clarity, and grammatical correctness. Please provide better link of this paragraph to your previous text.

Line 95 – Why did you opt for Mannitol Salt Agar as an isolation medium for Staphylococcus aureus? Baird-Parker Agar (especially the newer variants that allow for immediate detection of coagulase) is much more reliable.

Line 126 (and throughout the whole text) – please correct “para acetic” to “peracetic”.

Line 134 – the plural of “medium” is “media”; please correct accordingly.

Line 138 – 139: Please consider replacing “the restriction of growth” with “growth inhibition”.

Line 145 – “was realized as growth in Nutrient medium in the presence of 200 ppm”. There are several things to correct in this fragment. “Realized” is not the correct verb in this context – it means to become aware. Please use, for instance, “performed” instead of "realized”. “Nutrient medium” – please specify if Nutrient Broth or Nutrient Agar was used in your experiment.

Line 148 – Please see comment to line 145 regarding the use of the verbal form “realized”.

Line 150 – “mediums” is not the correct plural form of medium. Please correct.

Line 153 – For the sake of coherence, please capitalize all initials in “Principal Components Analysis”.

Line 156 – 158: Please delete this fragment of the MDPI template.

Line 164 – 165: Please specify what do you mean by “high cell counts”; I think you mean to say “high colony counts”. Also, clarify what do you mean by “high” (number of colonies per plate).

Line 173: “The most common bacteria were the Citrobacter…” – Do you mean to say “the most commonly encountered bacterial species was Citrobacter…”? Please correct accordingly.

Lines 177 – 178: I honestly do not understand what you mean with “The Bacterial isolate IDs refers to the origin, commercial name and the milk type of artisanal and 177 industrial cheese samples.” Analyzing Table 1, the “bacterial isolate ID” column is some kind of laboratory code. I think this sentence is unnecessary and confusing. Please delete it.

Line 190 – I think you are mixing up concepts here. “High level of resistance” means that the bacteria are resistant to high doses of the antimicrobial compounds tested. I think you mean to say, instead, “broad range of resistance”, meaning that the bacteria are resistant to a high number of antibiotics. Please correct accordingly.

Lines 194 – 195: the sentence “Also it was shown … donors of antibiotic genes” is unintelligible. I honestly could not understand it. Please rewrite in a clear, grammatically correct manner.

Lines 180 – 214: The description of which bacterial species is resistant to which antibiotics makes the text tedious to read. Either the authors consider that this information is important enough to warrant presentation as a table or figure, making the text description more succinct, or they must reformulate the text to make it more fluent to read.

Line 199 – 200: “The multiple antibiotic resistance (MAR) index was used to define the high risk origin of bacteria” – another unintelligible sentence. How does the MAR index tell you the origin of the bacteria? Please rewrite this sentence, using grammatically correct English.

Lines 202 – 203: “The MAR index value higher than 0.2 indicates 202 that the bacterial strains source is where antibiotics are often used” – please provide scientific evidence that this index does, indeed, show whether the bacteria come from an environment where they are frequently exposed to antibiotics. If no evidence (e.g., a citation) can be provided, this kind of reasoning should not be used in the manuscript.

Line 209 – “Principal component analysis (PCA) Figure 1” – Grammatically incorrect; please correct.

Line 224 – “expressed with” – incorrect adverb; please correct.

Lines 225 – 226: “Based on the results … function of bacterial strains” – this sentence is not correctly built; please correct grammar.

Line 226 – “The higher efficiency” is a comparative; I think you mean “The highest efficiency”. Please correct accordingly.

Line 238 – Incorrect use of “realized”; see previous comments on this issue.

Line 242 – I honestly do not understand what the authors mean by “significant positive correlation between antibiotics”. Please rewrite this using clear, grammatically correct language. Also, when stating that a correlation is significant, you need to indicate the p value obtained, or state (p<…).

Figure 3 – What exactly did you correlate in this figure? The diameters of the inhibition zones obtained when assessing antimicrobial activity of biocides and antibiotics? Please clarify, both in the text (line 242) and in the figure caption. If inhibition zones were used, also state the measuring units (mm?). Also, how many replicates for each bacterium and each antibiotic did you make? Please clarify this in the relevant part of the Materials and Methods Section.

I have serious qualms about the validity of correlating inhibition diameters in the kind of test you used. Agar diffusion tests have an inherent bias – the inhibition diameters obtained derive not only from the antimicrobial activity of the sample, but also from its capacity to diffuse across the agar under the test conditions. If the authors wanted to test for correlations, they should have assayed MICs in broth, which do not suffer from the previously mentioned bias. I think the authors should refrain from using correlation in this case, and base their discussion on PCA results only.

Line 253 – “media”, not “mediums”.

Table 2 – Please rewrite the heading. It is not correct from the grammatical point of view. This table is confusing. Why is the difference between MIC and “bacterial growth permitting concentration” so large? It seems, from these results, that the intervals between the concentrations tested were too broad and, therefore, a good estimate of the MICs was not obtained. This being the case, I would omit the MIC column and leave only the second column, renamed “lowest concentration permitting bacterial growth”.

Lines 260 – 272: The table covers the text in these lines, and this makes it impossible to provide a complete revision of the manuscript.

Line 273 – “survival of growth” (???) Please rephrase. Survival and growth are two different concepts. In this case, the test methodology was aimed at assessing survival, not growth.

Line 273 – “bacteria” (plural) instead of “bacterium” (singular).

Lines 275 – 280: Please rephrase and make this part of the text much more concise.

Figure 4 – Please make the legend more complete. What are the values shown? Diameter or radius of the inhibition zones? Which measuring unit was used (mm)?

Lines 155 – 319 – the whole Results section is a long, tedious, hard to read block of text. It needs to be way more succinct. I think the manuscript would benefit from combining the Results and the Discussion sections in a single Results and Discussion section.

Line 326 – “Antibiotic-resistant islands containing…” – please use precise language. E.g., “Pathogenicity islands containing genetic determinants of antibiotic resistance…”. Genome islands are not resistant to antibiotics – they confer resistance to the bacteria that harbour them.

Line 343 - It is the first time you mention C. braakii. Please give its name in full.

Line 344 – It is the first time you mention H. alvei. Please give its name in full.

Line 337 – Yarrowia lipolytica is a yeast – not relevant to the purpose of this manuscript. Please refer only to Kocuria.

Lines 349 – 350: “sulbactam sparfloxacin” – do you mean here a combination of both of these antibiotics? If yes, please write it correctly (e.g., sulbactam-sparfloxacin).

Lines 320 – 435: After reading the Discussion section, I highly recommend to present just one section, Results and Discussion, instead of two separate ones. The Discussion section, as it presently is, seems too disconnected from the Results. Both sections need to be completely rewritten, made more succinct, more streamlined, and more coherent. Also, attention must be paid to proper use of English. A revision by a native speaker is a must.

Lines 436 – 448 – Conclusions section – The conclusions should reflect the aims of the manuscript, and convey the main lines of the obtained results. From your conclusions, it is not clear whether the aim of determining the impact of the most commonly used biocides on antibiotic resistance was or not addressed. In my opinion, both sections need rewriting. The statement on the aims is to vague, making it hard to attain. It needs to be more detailed and more adequate to what the authors really made and achieved. The Conclusions section needs to be rewritten for clarity, to provide a better fit to the aims of the study and, also, to improve language.

Lines 450 – 451: Please remove these lines. Table/figure captions are not supplementary material. They need to be part of your manuscript.

A thorough revision of the text by a native English speaker is mandatory – the text is rife with grammar faults, making many sentences barely intelligible/unintelligible.

Round 2

Reviewer 1 Report

The authors have addressed most of my comments. I still have a couple of remaining questions:

1. What is the specific region of study?

2. How is Table 1 ordered? Currently, it appears that cheese types and bacterial types are listed without a clear organization. Additional details could be included to differentiate the cheese types from sources of isolation.

3. The isolation process remains unclear to me:

1) How is a “high colony count” determined for an isolate on selective media? Were colonies with similar morphology considered to be the same bacteria?

2) Were all colonies recovered on the agar sequenced?

3) Were all types of cheese plated on all types of agar plates? If these selective agar plates also supported the growth of other bacteria, were some of the identified bacteria expected to be found on multiple types of agar from one source of cheese?

4. In line 172, how is “most commonly encountered bacterial species” defined? It looks like that E. coli has been identified in the highest number of cheese samples.

5. For Figure 5 and 6, it would be helpful if the legends provided more information, such as explaining the abbreviations on the x-axis and the bolded columns.

Reviewer 2 Report

Authors have addressed all the comments. 

Author Response

Dear Reviewer,

We would also like to express our thanks for the positive feedback and helpful comments which helped us in improving greatly the quality of the manuscript.